# Effect of *Vibrio*-Derived Extracellular Protease vEP-45 on the Blood Complement System

**DOI:** 10.3390/biology10080798

**Published:** 2021-08-18

**Authors:** So Hyun Kwon, Jung Eun Park, Yeong Hee Cho, Jung Sup Lee

**Affiliations:** 1Department of Biomedical Science, College of Natural Sciences and Public Health and Safety, Chosun University, Gwangju 61452, Korea; hi6620@naver.com (S.H.K.); jepark@chosun.ac.kr (J.E.P.); choyh5181@gmail.com (Y.H.C.); 2Department of Integrative Biological Sciences & BK21 FOUR Educational Research Group for Age-Associated Disorder Control Technology, Chosun University, Gwangju 61452, Korea

**Keywords:** *Vibrio vulnificus*, vEP-45 protease, complement system, C3a, C5a, innate immunity

## Abstract

**Simple Summary:**

An extracellular *Vibrio* protease called vEP-45 can activate the complement system through an alternative pathway by cleaving key complement precursors, including C3 and C5, to their active forms, which can induce innate immunity.

**Abstract:**

*Vibrio vulnificus* is a pathogenic bacterium that can causes wound infections and fetal septicemia. We have reported that *V. vulnificus* ATCC29307 produces an extracellular zinc-metalloprotease (named vEP-45). Our previous results showed that vEP-45 can convert prothrombin to active thrombin and also activate the plasma kallikrein/kinin system. In this study, the effect of vEP-45 on the activation of the complement system was examined. We found that vEP-45 could proteolytically convert the key complement precursor molecules, including C3, C4, and C5, to their corresponding active forms (e.g., C3a, C3b, C4a, C4b, and C5a) in vitro cleavage assays. C5b production from C5 cleavage mediated by vEP-45 was not observed, whereas the level of C5a was increased in a dose-dependent manner compared to that of the non-treated control. The cleavage of the complement proteins in human plasma by vEP-45 was also confirmed via Western blotting. Furthermore, vEP-45 could convert C3 and C5 to active C3a and C5a as a proinflammatory mediator, while no cleavage of C4 was observed. These results suggest that vEP-45 can activate the complement system involved in innate immunity through an alternative pathway.

## 1. Introduction

The coagulation and blood complement systems are closely related in order to maintain blood homeostasis [1,2]. The blood complement system plays a crucial role in the innate immune response against common pathogens [3,4]. The system is composed of plasma proteins produced mainly by the liver or membrane proteins expressed on the cell surface [5]. The complement proteins collaborate in a cascade to opsonize pathogens and induce a series of inflammatory responses to assist immune cells in controlling infections and maintaining homeostasis [6]. The complement cascade can be activated by three pathways including the classical, the lectin pathways, and the alternative [5]. Each activation pathway leads to the generation of the C3 and C5 convertase enzyme complexes [7]. The binding of C5b (104 kDa) to a target cell initiates membrane attack complex (MAC) formation and cell lysis. Opsonins promote the phagocytic uptake of pathogens by scavenger cells [5,8]. The complement anaphylatoxins C3a (9 kDa) and C5a (11 kDa) contribute to inflammation and activate immune cells such as neutrophils, monocytes, and mast cells, which express the G-protein coupled anaphylatoxin receptors C3aR and C5aR [9,10]. C3a is implicated in the adaptive immunity by eliciting a monoclonal response from B cells and upregulating pro-inflammatory cytokines such as tumor necrosis factor alpha (TNF-α), interleukin 1 beta (IL-1β), and IL-6. In the presence of C5L2, C5a mediates C5aR internalization, and subsequent induction of extracellular signal-regulated kinase signaling and pro-inflammatory activation of macrophages [11].

Proteins secreted by various pathogens have been found to regulate the immune response. *Vibrio vulnificus* is a Gram-negative marine bacterium that causes wound infections and septicemia. *V. vulnificus*-derived proteases have been reported to show various biological functions: proteolytic degradation of a wide variety of host proteins such as plasma proteins involved in coagulation functions, induction of hemorrhagic tissue damage, and enhancement of vascular permeability via generation of inflammatory mediators [12]. We have previously shown that *V. vulnificus* secretes a zinc metalloprotease, named vEP-45, that interferes with blood homeostasis via prothrombin activation and fibrinolysis [13], and can also activate the plasma contact system by cleaving key zymogen molecules, thereby participating in the intrinsic pathway of coagulation and the kallikrein/kinin system [14]. However, it is not known whether vEP-45 can activate the blood complement system.

In this study, we evaluated the effect of vEP-45 protease on the activation of the complement system in vitro and in human plasma. Our results demonstrate that vEP-45 can activate the blood complement system by cleaving C3 and C5 to yield their active forms, which may consequently enhance innate immunity.

## 2. Materials and Methods

### 2.1. Materials

Purified human complement proteins, including C3, C3a, C3b, C4, C4a, C4b, C5, C5a, and C5b, and various monoclonal antibodies against C3, C3a, C4, C4a, C5, and C5a, were purchased from Complement Technology (Tyler, TX, USA). Sodium dodecyl sulphate (SDS), 1,10-phenanthroline (1,10-PT), tetramethylethylenediamine (TEMED), and other chemicals were purchased from Sigma-Aldrich (St. Louis, MO, USA). Human plasma was prepared as described previously [12].

### 2.2. Expression and Purification of vEP-45 Protease

*E. coli* DH5α cells were cultured in Luria Bertani medium. The vEP-45 protease was expressed and purified from *E. coli* DH5α cells harboring a recombinant plasmid pvEP-45 as described previously [13].

### 2.3. SDS-Polyacrylamide Gel Electrophoresis (SDS-PAGE) and Western Blot Analysis

SDS-PAGE was performed according to the method described by Laemmli [15]. Protein samples were mixed with an equal volume of 6× SDS-PAGE sample buffer, boiled at 100 °C for 3 min, and then loaded onto 10% or 15% polyacrylamide gel. Subsequently, SDS-PAGE and Western blot analysis were performed in the same method as mentioned in the previous paper [16]. The specific primary antibodies (C3, C3a, C4, C4a, C5, and C5a) were diluted 1:5000 in blocking buffer for overnight at 4 °C, and horseradish peroxidase-conjugated secondary antibodies (1:4000 in the blocking buffer) at 20 °C for 2 h. The band intensities were quantified using the ImageJ 1.52a program (National Institutes of Health, Bethesda, MD, USA).

### 2.4. Cleavage of C3, C4, and C5 by vEP-45 In Vitro

Reaction mixtures consisting of 5 µg of human complement proteins and different concentrations of vEP-45 (0.5, 2, 10, or 50 ng) in phosphate-buffered saline (PBS, pH 7.5) were incubated for 10 min at 37 °C. When Western blotting was performed, reaction mixtures consisting of human complement proteins C3 (1 µg), C4 (0.5 µg), or C5 (0.5 µg), and vEP-45 (0, 10, 25, 50, 100, 200, or 300 ng) in PBS (pH 7.5) were incubated for 3 min at 37 °C. Thereafter, the reactions were terminated via addition of 1 mM 1,10-PT as described previously [14]. The cleaved products were separated via 10% or 15% SDS-PAGE and visualized via Coomassie blue staining or Western blotting.

### 2.5. Cleavage of C3 and C5 by vEP-45 in Human Plasma

Human plasma was diluted using PBS (pH 7.5) to a final concentration of 10%. Reaction mixtures consisting of 10 µL of 10% human plasma and vEP-45 (0.5, 1, or 2 µg) diluted in PBS (pH 7.5) were incubated for 3 min at 37 °C. Thereafter, the reactions were terminated via addition of 1 mM of 1,10-PT as described previously [14]. The cleaved products were separated via SDS-PAGE and detected via Western blotting.

## 3. Results

### 3.1. Cleavage of C3 by vEP-45

The cleavage of the human complement protein C3 by vEP-45 into C3a and C3b was evaluated in vitro (Figure 1A–C and Figure 2A–C). When C3 (5 µg) and vEP-45 (0.5, 2, 10, or 50 ng) were incubated for 10 min at 37 °C and analyzed via SDS-PAGE, bands of sizes 101 kDa and 9 kDa were produced from the α′ chain of C3b and C3a, respectively (Figure 1A). Maximal production of C3b and C3a from C3 was the observed when 0.5 ng of vEP-45 treatment was used, accompanied with a gradual decrease at higher vEP-45 concentrations (Figure 1B,C). To determine whether a similar in vitro cleavage pattern could be detected via Western blot analysis, C3 (1 µg) and vEP-45 (0, 25, 50, 100, 200, or 300 ng) were incubated for 3 min at 37 °C and analyzed via Western blotting. C3b and C3a were produced in a dose-dependent manner from the precursor C3 via vEP-45-mediated cleavage (Figure 2A–C). These results suggest that vEP-45 may proteolytically cleave C3 into C3a and C3b.

### 3.2. Cleavage of C4 by vEP-45 In Vitro

The ability of vEP-45 to cleave the human complement protein C4 into C4a and C4b was evaluated in vitro (Figure 1D–F and Figure 2D–F). When protein C4 (5 µg) and vEP-45 (0.5, 2, 10, or 50 ng) were incubated for 10 min at 37 °C and subjected to SDS-PAGE, C4b α′ chain and C4a formed two bands of sizes 88 kDa and 9 kDa, respectively (Figure 1D). Similar to C3 cleavage, maximal C4b production was obtained using 0.5 ng of vEP-45 and production decreased at higher concentrations; however, C4a production peaked at a treatment of 50 ng of vEP-45 in a dose-dependent manner (Figure 1E,F). To determine whether a similar in vitro cleavage pattern could be obtained via Western blotting, C4 (0.5 µg) and vEP-45 (0, 10, 25, 50, 100, 200, or 300 ng) were incubated for 3 min at 37 °C and analyzed via Western blotting. Maximal production of C4b and C4a from C4 was obtained by using 50 and 200 ng of vEP-45, respectively, and was accompanied with a gradual decrease at higher concentrations (Figure 2D–F). Therefore, vEP-45 may proteolytically cleave C4 into C4a and C4b.

### 3.3. Cleavage of C5 by vEP-45 In Vitro

The ability of vEP-45 to cleave the human complement protein C5 into C5a and C5b was evaluated in vitro (Figure 1G–I and Figure 2G–I). When C5 (5 µg) and vEP-45 (0.5, 2, 10, or 50 ng) were incubated for 10 min at 37 °C and analyzed via SDS-PAGE, C5a produced a 11 kDa band whereas the band of the α′ chain of C5b was barely visible (Figure 1G). The generation of C5a by vEP-45 increased in a dose-dependent manner (Figure 1I). To determine whether a similar in *vitro* cleavage pattern could be detected via Western blotting analysis, C5 (0.5 µg) and vEP-45 (0, 10, 25, 50, 100, 200, or 300 ng) were incubated for 3 min at 37 °C and then analyzed via Western blotting. C5a was produced from C5 via vEP-45-mediated cleavage in a dose-dependent manner (Figure 2G–I); however, the band of the α′ chain of C5b was not detected in the blot. These results suggest that vEP-45 may proteolytically cleave C5 into C5a.

### 3.4. Activation of the Complement System in Human Plasma by vEP-45

The aforementioned in vitro experiments showed that vEP-45 could convert the complement proteins into their active forms (Figure 1 and Figure 2). Therefore, we evaluated whether vEP-45 can activate these complement proteins present in plasma. Human plasma (10%) was treated with vEP-45 (0.5, 1, or 2 µg) for 3 min at 37 °C and subjected to Western blot analysis. The vEP-45-mediated production of C3b and C3a increased to approximately 25.6% and 55%, respectively, in a dose-dependent manner (Figure 3A–C). Similar to the in vitro findings, the band of the α′ chain of C5b was barely visible, whereas the production of C5a increased via vEP-45 treatment to approximately 43.7% in a dose-dependent manner, compared to that of the non-treated control (Figure 3D–F). These results suggest that vEP-45 may convert C3 and C5 to their respective active factors, even in human plasma. C3 and C5 were converted into functional fragments which was consistent with the in vitro results; however, cleavage of C4 was not observed.

## 4. Discussion

The present study analyzed the changes in blood homeostasis by investigating the activation of the blood complementary system by vEP-45 derived from a *Vibrio* strain isolated from a *Vibrio* infection. A summary of the complement system activation is depicted in Figure 4. The blood complement system is a part of the immune system that enhances the activity of antibodies and phagocytic cells to remove microbes and damaged cells, promote inflammation, and attack the cell membrane of pathogens [4,17,18]. Therefore, activation of the blood complement system by vEP-45 during *Vibrio* infections is crucial in the innate immune system.

The complement system can be activated by three pathways (the classical, the lectin, and the alternative pathways), resulting in cleavage of inactive C3 protein into functional fragments C3a and C3b [6,19]. The C3 precursor is composed of an α chain (110 kDa) and β chain (75 kDa) that are linked by an intrastrand disulfide bond [20]. The cleavage of C3 at the α chain leads to the generation of C3b, which comprises two polypeptide chains (101 kDa α′ chain and 75 kDa β chain) and C3a (9 kDa) (Appendix A) [20]. We found that C3a and C3b produced via vEP-45 treatment increased in a dose-dependent manner (Figure 1A,C), which was also observed in human plasma (Figure 3). These results suggest that the functional fragment C3b may induce the subsequent cleavage of C5, and C3a may function as a proinflammatory mediator.

The active complement C3b binds to C3 convertase to form a new enzymatic complex, C5 convertase, which cleaves C5 into anaphylatoxins C5a and C5b [3]. The cleavage of C5 at the α chain releases C5b which comprises two polypeptide chains (104 kDa α′ chain and 75 kDa β chain) and C5a (11 kDa) (Appendix A) [21]. C5b is the first complement component to initiate MAC formation which mediates cell lysis. The other fragment C5a performs functions that lead to the recruitment of immune cells, which activate phagocytic cells and mediate the release of granule-based enzymes and generation of oxidants, all of which may contribute to innate immune functions or tissue damage [22,23]. C5a provides a vital bridge between the innate and adaptive immune functions, extending the roles of C5a in inflammation [22,24]. In our study, C5 was found to be converted to C5a and C5b via vEP-45 treatment both in vitro and in human plasma (Figure 1 and Figure 3). These results suggest that the generated C5b functions as an initiator of MAC formation, and that C5a may induce an inflammatory response.

C4 originates from the human leukocyte antigen system and plays critical roles in immunity, tolerance, and autoimmunity [25]. Both classical and lectin pathways involve the cleavage of C4 into active fragment C4b and a small fragment C4a [26]. The complement C4 precursor is composed of three chains, the α chain (97 kDa), β chain (75 kDa), and γ chain (33 kDa), which are linked by intrastrand disulfide bonds [27]. The cleavage of C4 at the α chain releases C4b which comprises three polypeptide chains (88 kDa α′ chain, 75 kDa β chain, and 33 kDa γ chain) and C4a (9 kDa) (Appendix A) [27]. In this study, the conversion of C4 into C4a and C4b via vEP-45 treatment was observed in vitro; however, the production of C4a and C4b was not detected in human plasma. These results suggest that the activation of the complement system by vEP-45 may occur via the alternative pathway among the three pathways.

In a previous study, the activation of the contact system by vEP-45 has been analyzed [14], whereas in this study, the activation of the complement system was observed. Brady kinin (BK), C3a, and C5a generated as a result of activation of the two systems (contact and complement systems) are well-known mediators for inducing inflammation [28,29,30], which suggests the possibility of inducing an inflammatory response. Therefore, future studies should evaluate the effects of vEP-45 on the innate immune response in the human body associated with the pathogen-associated molecular patterns during *Vibrio* infections.

## 5. Conclusions

Our data show that (1) vEP-45 can cleave the blood complement factors (C3, C4, and C5) to convert to the active forms in vitro and (2) the protease can also cleave C3 and C5 to generate the active forms in plasma milieu. Therefore, we conclude that vEP-45 can activate the complement system through an alternative pathway and consequently enhance innate immunity.

## Figures and Tables

**Figure 1 biology-10-00798-f001:**
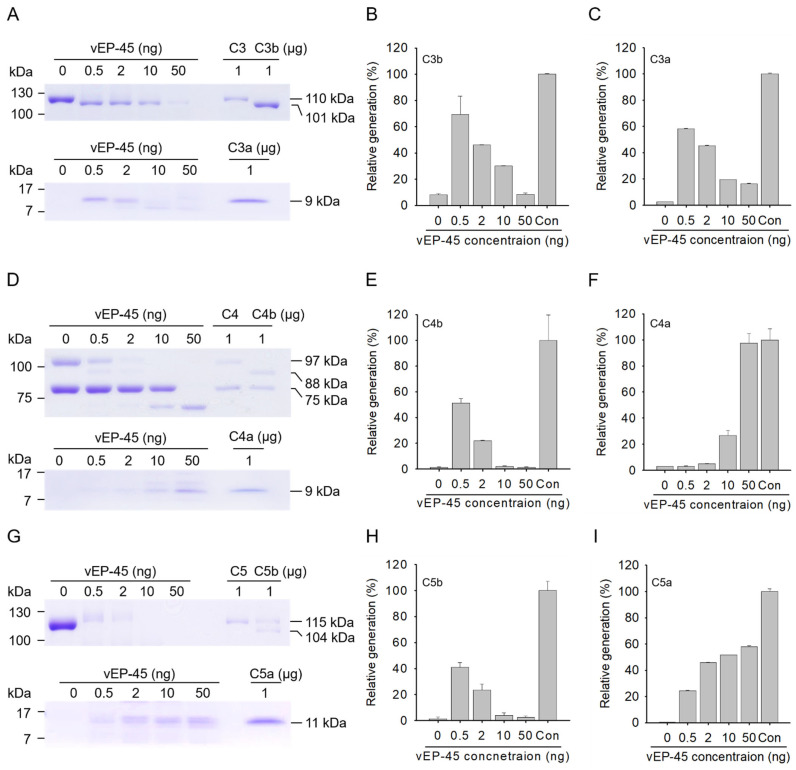
Cleavage of human complement proteins (C3, C4, and C5) by vEP-45 as determined by SDS-PAGE. (**A**,**D**,**G**) Human complement protein C3, C4, or C5 (each 5 µg) was incubated with vEP-45 (0.5, 2, 10, or 50 ng) for 10 min at 37 °C. Proteins from each sample were separated via SDS-PAGE and stained with Coomassie brilliant blue. The histograms show the production of (**B**) C3b α′ chain, (**C**) C3a, (**E**) C4b α′ chain, (**F**) C4a, (**H**) C5b α′ chain, and (**I**) C5a upon treatment with different concentrations of vEP-45, in which the values obtained using the control (Con; only C3b, C3a, C4b, C4a, C5b, and C5a) were considered as 100%. The uncropped SDS-polyacrylamide gel images can be found in Appendix A.

**Figure 2 biology-10-00798-f002:**
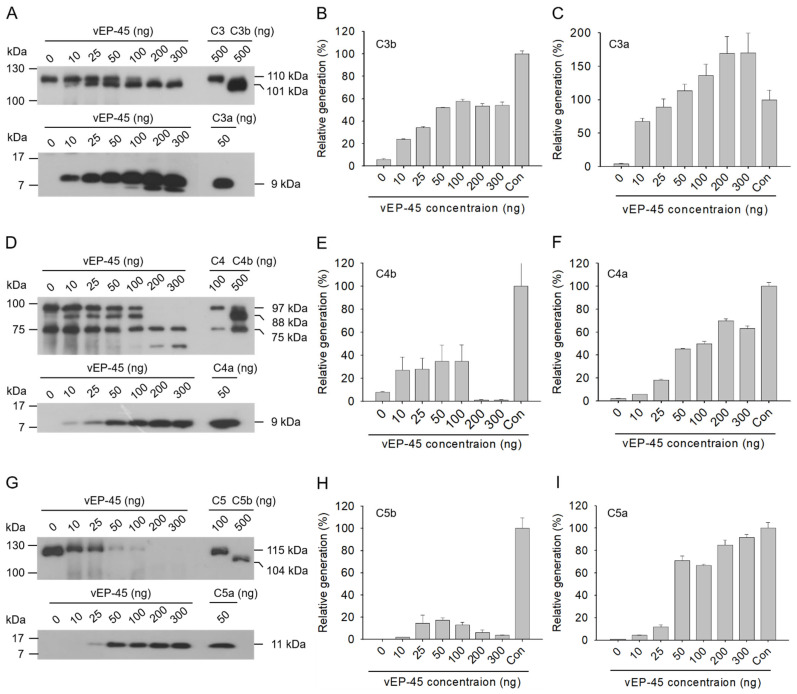
Cleavage of human complement proteins by vEP-45 as detected with Western blotting. (**A**,**D**,**G**) Human complement protein C3 (1 µg), C4 (0.5 µg), or C5 (0.5 µg) was incubated with vEP-45 (10, 25, 50, 100, 200, or 300 ng) for 3 min at 37 °C. Proteins from each sample were separated by SDS-PAGE; Western blotting was performed using anti-C3 antibody (**A** upper panel), anti-C3a antibody (**A** lower panel), anti-C4 antibody (**D** upper panel), anti-C4a antibody (**D** lower panel), anti-C5 antibody (**G** upper panel), or anti-C5a antibody (**G** lower panel). The histograms show the production of (**B**) C3b α′ chain, (**C**) C3a, (**E**) C4b α′ chain, (**F**) C4a, (**H**) C5b α′ chain, and (**I**) C5a upon treatment with different concentrations of vEP-45, in which the values obtained using control (Con; only C3b, C3a, C4b, C4a, C5b, and C5a) were regarded as 100%. The uncropped Western blot images can be found in Appendix A.

**Figure 3 biology-10-00798-f003:**
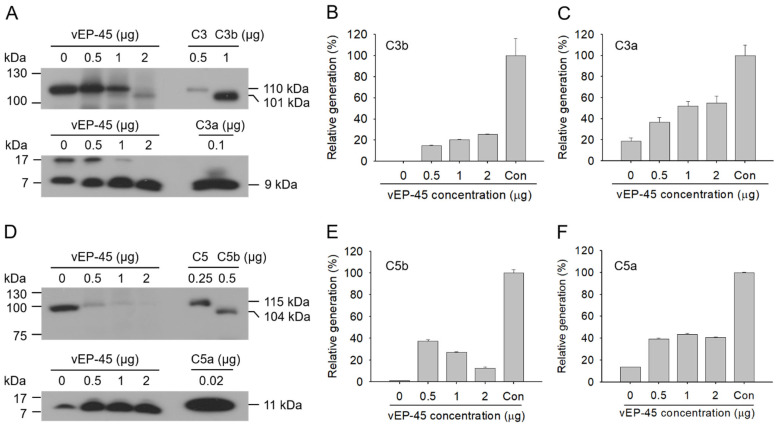
Cleavage of C3 and C5 by vEP-45 in human plasma. (**A**,**D**) Samples of human blood plasma (10%) were incubated with vEP-45 (0.5, 1, or 2 µg) for 3 min at 37 °C. Proteins from each sample were separated by SDS-PAGE; Western blotting was performed using anti-C3 antibody (**A** upper panel), anti-C3a antibody (**A** lower panel), anti-C5 antibody (**D** upper panel), or anti-C5a antibody (**D** lower panel). The histograms show the production of (**B**) C3b α′ chain, (**C**) C3a, (**E**) C5b α′ chain, and (**F**) C5a upon treatment with different concentrations of vEP-45, in which the values obtained from a control (Con; only C3b, C3a, C5b, and C5a) were considered as 100%. The uncropped Western Blot images can be found in Appendix A.

**Figure 4 biology-10-00798-f004:**
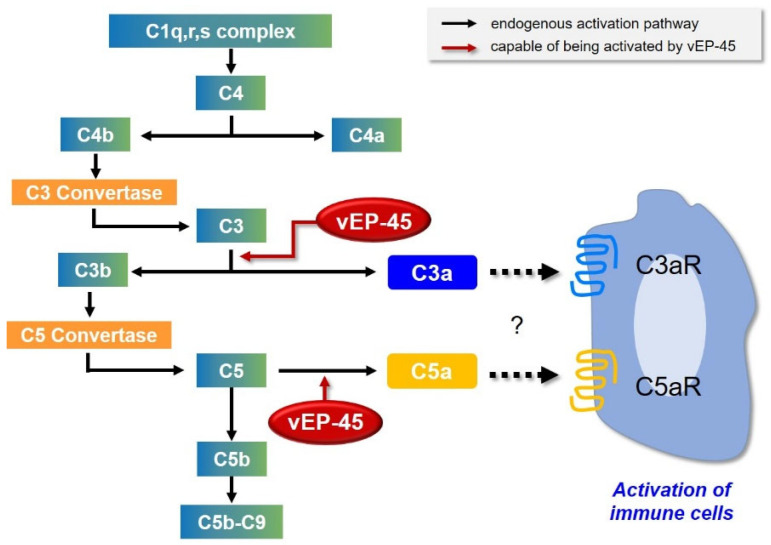
Effects of vEP-45 on the complement system during *Vibrio* infection.

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
