# Peer review of "Effect of Vibrio-Derived Extracellular Protease vEP-45 on the Blood Complement System"

_biology, 2021, doi:10.3390/biology10080798_

Round 1

Reviewer 1 Report

Dear Authors,

The authors investigated the activation of the complement system by the vEP-45 effect.

The manuscript needs more experiments, preferably using different blotting techniques.

Author Response

Thank you for the valuable comment. However, based on our overall strategy, we ask for your understanding. We are analyzing the effect of vEP-45 on the innate immunity response during Vibrio infection. The innate immune response is induced by 1) complement system activation, 2) inflammatory response, and 3) recruitment of immune cells. As the components of the complement system are the first to encounter an antigen, we focused on the complement system in this study. In future studies, we will analyze the inflammatory response and immune cell recruitment” for better clarity and readability.

Reviewer 2 Report

In the manuscript "Activation of the blood complement system by a Vibrio-derived extracellular protease vEP-45", Kwon, Park, et al. demonstrate that extracellular protease vEP-45 encoded by V. vulnificus can cleave proteins of the complement system: C3, C4, and C5.

The manuscript is clear and well written, with conclusions corresponding to the presented results. However, there are several points which the authors should address before publication.

1) My primary concern is the authors' approach for presenting data in the bar charts in Figures 1-3, with cleaved components of the complement system being normalized to 100% in the absence of the protease, produces artifacts. For example, in Figure 2H, the bar chart shows a 3-fold increase in the generation of C5b, although the signal comes from nonspecific degradation of the C5, as the authors themselves admit in line 162. While such an approach would be valid for quantifying the non-cleaved component, it is not for the cleaved components, which should not be present in significant amounts until the protease is added. Thus, without protease, any signal corresponding to the cleaved complement system should be treated as a background and represented with 0%. The intensity of the 100% signal can be calculated from the control lanes on the right of each gel/membrane, after correcting for the difference in the amount of protein.

2) In addition to the cleavage of the tested complement proteins, the gels and membranes clearly show that protease vEP-45 also unspecifically degrades proteins. For example, in Figure 1G, in the first lane (no protease added), much more protein is detected than in other lanes, although the same amount of reaction mixture was loaded into each lane. This degradation can easily be quantified using bar charts, similarly as the authors quantified generation of the cleaved complement components. Following quantification, these results should be added next to the bar charts describing the accumulation of cleaved complement components. This approach would allow the reader to assess how specific is the activation of complement by the vEP-45.

Other minor concerns are:

3) The authors should specify exactly how they quantified the proteins (what software and what algorithms they used)

4) Some lanes in figures seem to be mislabelled: on the SDS-PAGE gel, C4a is mislabelled as C3a in Figure 1D; on the membranes in Figure 2, C3a is mislabelled as C3, C4a is mislabelled as C4, and C5a is mislabelled as C5; on the membranes in Figure 3, C3a is mislabelled as C3, and C5a is mislabelled as C5.

5) In line 172, it is not clear which measurement refers to 70%.

6) The authors should explain why they used much smaller quantities of vEP-45 protease in experiments in vitro than in human plasma.

7) Discussion should be supplemented by the biochemical mechanism behind physiological cleavage of the complement components (are they also cleaved by metalloproteinases?) and should address the question of whether vEP-45 cleaves proteins similarly.

8) Complement sizes listed in the Discussion should instead be presented in the Introduction to help to interpret the membranes.

9) I suggest putting into the abstract that based on the results of this study, V. vulnificus may function as a proinflammatory mediator.

Author Response

â–¶ Comment 1: My primary concern is the authors' approach for presenting data in the bar charts in Figures 1-3, with cleaved components of the complement system being normalized to 100% in the absence of the protease, produces artifacts. For example, in Figure 2H, the bar chart shows a 3-fold increase in the generation of C5b, although the signal comes from nonspecific degradation of the C5, as the authors themselves admit in line 162. While such an approach would be valid for quantifying the non-cleaved component, it is not for the cleaved components, which should not be present in significant amounts until the protease is added. Thus, without protease, any signal corresponding to the cleaved complement system should be treated as a background and represented with 0%. The intensity of the 100% signal can be calculated from the control lanes on the right of each gel/membrane, after correcting for the difference in the amount of protein. In addition to the cleavage of the tested complement proteins, the gels and membranes clearly show that protease vEP-45 also unspecifically degrades proteins. For example, in Figure 1G, in the first lane (no protease added), much more protein is detected than in other lanes, although the same amount of reaction mixture was loaded into each lane. This degradation can easily be quantified using bar charts, similarly as the authors quantified generation of the cleaved complement components. Following quantification, these results should be added next to the bar charts describing the accumulation of cleaved complement components. This approach would allow the reader to assess how specific is the activation of complement by the vEP-45.

â–¶ Response to comment 1: Thank you for your valuable comment. According to your suggestion, we calculated the control line in Figures 1-3 as 100% to show the relative generation (%) of active form of protein (C3b, C3a, C4b, C4a, C5b, and C5a) from inactive protein (C3, C4, and C5). We have added the modified figures in the revised manuscript.

----------------------------------------------------------------------------------------
â–¶ Comment 2: The authors should specify exactly how they quantified the proteins (what software and what algorithms they used

â–¶ Response to comment 2: According to your suggestion, we have mentioned the software used for quantification on lines 91-93. “The band intensity was quantified using the ImageJ software (National Institutes of Health, Bethesda, Maryland, USA).”

----------------------------------------------------------------------------------------
â–¶ Comment 3: Some lanes in figures seem to be mislabelled: on the SDS-PAGE gel, C4a is mislabelled as C3a in Figure 1D; on the membranes in Figure 2, C3a is mislabelled as C3, C4a is mislabelled as C4, and C5a is mislabelled as C5; on the membranes in Figure 3, C3a is mislabelled as C3, and C5a is mislabelled as C5.

â–¶ Response to comment 3: Thank you for kind comments. We have corrected the mislabeled in figures and have modified the Figures 1, 2, and 3.

----------------------------------------------------------------------------------------
â–¶ Comment 4: In line 172, it is not clear which measurement refers to 70%.

â–¶ Response to comment 4: We apologize for the unclear text. We have made the changes as per your comment. The changes have been indicated in red.

----------------------------------------------------------------------------------------
â–¶ Comment 6: The authors should explain why they used much smaller quantities of vEP-45 protease in experiments in vitro than in human plasma.

â–¶ Response to comment 6: Since there is a difference in the reactivity of vEP-45, the concentration was adjusted to improve the band visibility of the activated form.

----------------------------------------------------------------------------------------
â–¶ Comment 7: Discussion should be supplemented by the biochemical mechanism behind physiological cleavage of the complement components (are they also cleaved by metalloproteinases?) and should address the question of whether vEP-45 cleaves proteins similarly.

â–¶ Response to comment 7: Thank you for the suggestion. Based on the suggestion, we have showed the biochemical mechanism underlying the physiological cleavage of the complement components in the Supplementary Figure 1.

----------------------------------------------------------------------------------------

â–¶ Comment 8: Complement sizes listed in the Discussion should instead be presented in the Introduction to help to interpret the membranes.

â–¶ Response to comment 8: We have provided the complement sizes in the introduction and Supplementary Figure 1.

----------------------------------------------------------------------------------------

â–¶ Comment 9: I suggest putting into the abstract that based on the results of this study, V. vulnificus may function as a proinflammatory mediator.

â–¶ Response to comment 9: According to your suggestion, we have modified the sentence in abstract. The changes have been indicated in red.

Reviewer 3 Report

The authors claim to have identified new putative substrates of the secreted metalloprotease vEP-45 of the Vibrio vulnificus, a related bacterial pathogen to Vibrio cholerae. In vitro work shows that the C3, C4, and C5 of the complement cascade are potentially substrates of vEP-45. Cleavage in presence of human plasma shows only C3 and C5 can be processed. Because the cleaved products resemble the immunomodulatory molecules C3a and C5a, the authors claim that vEP-45 can generate an immune response by activating the complement cascade.

-------------------------------------------------------------

Comments and suggestions for Authors:

line 2 Although the authors are providing some evidence of potential processing of C3 C4 and C5, it is unclear where exactly the putative cleavage of the substrates would be occurring. Without this information, we cannot be sure about the nature of the product and how to compare it to the known immunomodulatory forms, C3a and C5a. This has consequences throughout the paper wherever authors claim that the protease is generating immunologically active molecules. In fact, the paper lacks any direct evidence that the product of cleavage of the substrates are yielding immunologically active molecules. Therefore I suggest to change the title and related statements accordingly, and use a more prudent language on the matter, unless the authors do provide experimental evidence that the various products of vEP-45 processing are directly immunomodulatory.

line 39 Please explain in the introduction part, the literature around the conversion of C3, C4, C5 into the various forms.

Figure 1 A The cleavage of the full length C3 is yielding bands that are similar but not necessarily matching with C3a, especially at higher protease amounts (10, 50). Please describe these bands thoroughly.

Figure 3 D similarly the cleavage of C5 in the plasma produces a band that is smaller than the 11 kDa band. Please comment this band.

To reorganise the figures 1 and 2 into 3 figures such that the figures focus on the each putative substrates and the flow follows that one of the text.

Please represent Fig 1 B C E F H I Fig 2 B C E F H and I, relative to the input, and not relative to the background of the no-enzyme-treated control.

Please explain how quantifications and statistics of gels and western blots were performed to obtain the bar graphs and their error bars.

Please label thoroughly the supplementary figures.

Author Response

â–¶ Comment 1: line 2 Although the authors are providing some evidence of potential processing of C3 C4 and C5, it is unclear where exactly the putative cleavage of the substrates would be occurring. Without this information, we cannot be sure about the nature of the product and how to compare it to the known immunomodulatory forms, C3a and C5a. This has consequences throughout the paper wherever authors claim that the protease is generating immunologically active molecules. In fact, the paper lacks any direct evidence that the product of cleavage of the substrates are yielding immunologically active molecules. Therefore I suggest to change the title and related statements accordingly, and use a more prudent language on the matter, unless the authors do provide experimental evidence that the various products of vEP-45 processing are directly immunomodulatory.

â–¶ Response to comment 1: Thank you for the suggestions. Accordingly, we have showed the biochemical mechanism underlying physiological cleavage of the complement components in Supplementary Figure 1. To compare it with the known immunodulatory forms, we also confirmed via western blotting using specific antibodies against C3a and C5a. Furthermore, to clarify this issue, we have revised the title from “Activation of the blood complement system by a Vibrio-derived extracellular proteases vEP-45” to “Effect of Vibrio-derived extracellular protease vEP-45 on the blood complement system”.

----------------------------------------------------------------------------------------
â–¶ Comment 2: line 39 please explain in the introduction part, the literature around the conversion of C3, C4, C5 into the various forms.

â–¶ Response to comment 2: Thank you for your valuable comment. We have explained the conversion of C3, C4, and C5 into various forms in introduction and Discussion sections.

----------------------------------------------------------------------------------------

â–¶ Comment 3: Figure 3 D similarly the cleavage of C5 in the plasma produces a band that is smaller than the 11 kDa band. Please comment this band.

â–¶ Response to comment 3: It was observed that the same phenomenon occurred in the case of human plasma in several experiments and in results shown in Figure 3A. Therefore, we believe that warpage occurred during electrophoresis due to differences in plasma samples or amount of proteins.

----------------------------------------------------------------------------------------

â–¶ Comment 4: To reorganise the figures 1 and 2 into 3 figures such that the figures focus on the each putative substrates and the flow follows that one of the text. Please represent Fig 1 B C E F H I Fig 2 B C E F H and I, relative to the input, and not relative to the background of the no-enzyme-treated control.

â–¶ Response to comment 4: According to your comment, we have changed the presentation of the results and displayed them as not based on the background of the no-enzyme-treated control.

----------------------------------------------------------------------------------------

â–¶ Comment 5: Please explain how quantifications and statistics of gels and western blots were performed to obtain the bar graphs and their error bars. Please label thoroughly the supplementary figures.

â–¶ Response to comment 5: Thank you for your valuable comment. We have mentioned the software used for quantification on lines 91-93. “The band intensity was quantified using ImageJ software (National Institutes of Health, Bethesda, Maryland, USA).”

Round 2

Reviewer 1 Report

Dear Authors, thanks for the clarifications.